# Seroprevalence of Chikungunya virus and living conditions in Feira de Santana, Bahia-Brazil

Maria Glória Teixeira[1]*, Lacita Menezes Skalinski[1,2], Enny S. Paixão[3], Maria da Conceição N. Costa[1], Florisneide Rodrigues Barreto[1], Gubio Soares Campos[4], Silvia Ines Sardi[4], Rejane Hughes Carvalho[4], Marcio Natividade[1], Martha Itaparica[1], Juarez Pereira Dias[1], Soraya Castro Trindade[5], Bárbara Pereira Teixeira[1], Vanessa Morato[6], Eloisa Bahia Santana[7], Cristina Borges Goes[8], Neuza Santos de Jesus Silva[7], Carlos Antonio de Souza Teles Santos[5,9], Laura C. Rodrigues[3], Jimmy Whitworth[3]

1 Instituto de Saúde Coletiva/ Universidade Federal da Bahia, Salvador-BA, Brazil, 2 Departamento de Ciências da Saúde/ Universidade Estadual de Santa Cruz, Ilhéus-BA, Brazil, 3 London School of Hygiene and Tropical Medicine, London, United Kingdom, 4 Instituto de Ciências da Saúde/ Universidade Federal da Bahia, Salvador-BA, Brazil, 5 Universidade Estadual de Feira de Santana, Feira de Santana-BA, Brazil, 6 Secretaria de Segurança Pública do Estado da Bahia, Salvador-BA, Brazil, 7 Secretaria Municipal de Saúde de Feira de Santana, Feira de Santana-BA, Brazil, 8 Centro Universitário UNIFTC de Feira de Santana, Feira de Santana-BA, Brazil, 9 Instituto Gonçalo Muniz/Fiocruz, Salvador-BA, Brazil

* magloria@ufba.br

**Data Availability Statement:** All relevant data are within the manuscript and its Supporting Information files.

## Abstract

### Background

Chikungunya is an arbovirus, transmitted by *Aedes* mosquitoes, which emerged in the Americas in 2013 and spread rapidly to almost every country on this continent. In Brazil, where the first cases were detected in 2014, it currently has reached all regions of this country and more than 900,000 cases were reported. The clinical spectrum of chikungunya ranges from an acute self-limiting form to disabling chronic forms. The purpose of this study was to estimate the seroprevalence of chikungunya infection in a large Brazilian city and investigate the association between viral circulation and living condition.

### Methodology/principal findings

We conducted a population-based ecological study in selected Sentinel Areas (SA) through household interviews and a serologic survey in 2016/2017. The sample was of 1,981 individuals randomly selected. The CHIKV seroprevalence was 22.1% (17.1 IgG, 2.3 IgM, and 1.4 IgG and IgM) and varied between SA from 2.0% to 70.5%. The seroprevalence was significantly lower in SA with high living conditions compared to SA with low living condition. There was a positive association between CHIKV seroprevalence and population density (r = 0.2389; p = 0.02033).

### Conclusions/significance

The seroprevalence in this city was 2.6 times lower than the 57% observed in a study conducted in the epicentre of the CHIKV epidemic of this same urban centre. So, the herd

**Funding:** This work was supported by three grants: Medical Research Council (MRC/UK - https://mrc.ukri.org/) under Grant MC/PC 15084 to JW, Coordenação de Aperfeiçoamento de Pessoal de Nível Superior (CAPES/BR - https://www.gov.br/capes/pt-br) under Grant 0451/2016 to MGT and Brazilian Ministry of Health (www.saude.gov.br) under Decentralized Execution Term 130/2017 UFBA/MS to MGT. ESP is funded by the Wellcome Trust (grant number 213589/Z/18/Z). The funders had no role in study design, data collection and analysis, decision to publish, or preparation of the manuscript.

**Competing interests:** The authors have declared that no competing interests exist.

immunity in this general population, after four years of circulation of this agent is relatively low. It indicates that CHIKV transmission may persist in that city, either in endemic form or in the form of a new epidemic, because the vector infestation is persistent. Besides, the significantly lower seroprevalences in SA of higher Living Condition suggest that beyond the surveillance of the disease, vector control and specific actions of basic sanitation, the reduction of the incidence of this infection also depends on the improvement of the general living conditions of the population.

## Author summary

Chikungunya is a disease caused by viruses transmitted by *Aedes* mosquitoes. It stands out because it can produce chronic cases and disabling forms. This survey was conducted in a large city in Brazil, involving 1,981 people living in different living conditions/LC, distributed in 30 urban areas. Interviews and blood collection were performed to know the seroprevalence of chikungunya virus (CHIKV). The global seroprevalence was 22.1%, ranged from 2.0 to 70.5%. The areas where the highest and lowest seroprevalence were of low LC, but population density was different among them, which was related to seroprevalence. The mosquito index was not associated with the seroprevalence. Our results showed that lower seroprevalence than another survey conducted in the epicentre of the epidemic of this same city. So, a considerable proportion of the population of Feira de Santana is susceptible to the CHIKV. Possibly the virus will continue to circulate and manifest itself as endemic or epidemic, as long as LC remain favourable to the reproduction of the transmitting vector of CHIKV.

## Introduction

Chikungunya has stood out among emerging arboviruses, because of its potential to produce epidemics with high rates of chronic and disabling sequeleae, especially arthritis, which can last from months to years [1]. The agent of this disease, chikungunya virus (CHIKV) belonging to *Togaviridae* family, is transmitted by mosquitoes of the genus *Aedes* [2]. Before 2004, human CHIKV infections were restricted to parts of Africa, and in 2005, a major outbreak occurred in islands of the Indian Ocean [3]. Since then, epidemics of chikungunya have been reported in Italy [4], India, Indonesia, Maldives, Myanmar and Thailand [5], and in 2013, chikungunya reached the Americas, where the first autochthonous cases were confirmed in the Caribbean island of St Martin[6], and quickly spread to over 43 territories, causing more than 2.6 million cases by 2017 [7].

There are already many studies about chikungunya seroprevalence, which have results ranging from 0.4% to over 75%. This variation is a reflection of sample selection (random or convenience), the magnitude of epidemics at each site, time of virus circulation before the survey, demographic and environmental characteristics of each site, among other aspects [8].

Previous studies on dengue, another arbovirus circulating in Brazil, have shown seroprevalence varying by place of residence, reflecting living conditions, such as overcrowding, low education, low income, poor sanitation and high exposure to the infected vectors [9,10]. Although CHIKV is transmitted by the same mosquito as dengue; there is a lack of information regarding the relationship between the occurrence of CHIKV infection and socio-economic conditions in the urban environment. Therefore, this study aims to estimate the

seroprevalence of chikungunya infection in a large Brazilian city and investigate the association between viral circulation and living condition, using an ecological approach.

## Methods

### Ethics statement

This research was approved by the Committee on Ethics in Research of Institute of Collective Health/Federal University of Bahia, under number 1.685.039/2016. Trained interviewers explained the project objectives, and, when allowed, collected participant's signatures in the Free and Informed Consent Form (FICF) and conducted the interviews. For participants <18 years old, we used the Term of Assent and a responsible adult signed the document.

### Design and local of study

We conducted a population-based ecological study through household interviews and serological survey from November 2016 to September 2017, involving residents of 30 delimited areas named "Sentinel Areas" (SA) in Feira de Santana (FSA) city, Northeast Brazil (Fig 1). This city is the second most populous in Bahia state, with 609,913 inhabitants, a population density of 416.03 inhabitants/ km$^2$, and it is 110 km far from Salvador, the capital of Bahia. Located in the transition zone between Zona da Mata (forest zone) and Sertão (dry forest), it has tropical weather with high temperatures and relative humidity ranging from 74 to 88% [11]. Chikungunya virus (CHIKV) serotype ECSA emerged in Brazil in 2014, in FSA [12] where it started to circulate intensely. In FSA, dengue epidemics has occurred for more than two decades and the ZIKV was identified in 2015 [13].

### Areas selection

To select the 30 SA we used the Living Conditions Index (LCI) as a proxy of living conditions (LC), similar to proposed by Paim et al (2003) [14]. For this, from the Demographic Census data of the year 2000, these authors used five indicators to the variables income (proportion of householders in permanent private households with average monthly income equal to or less than two minimum wages), slum (percentage of houses in subnormal agglomerate to the total

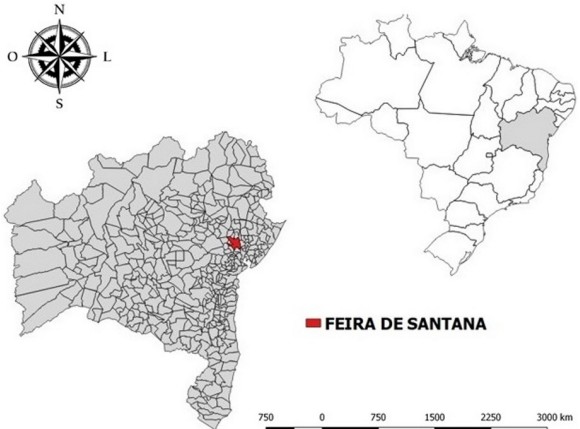

**Fig 1. Maps show locations of Feira de Santana in Bahia state and Bahia state in Brazil.** Map base layers were obtained from https://portaldemapas.ibge.gov.br/portal.php#mapa222663 and https://portaldemapas.ibge.gov.br/portal.php#mapa223096 covered by a Creative Commons Attribution 4.0 International (CC BY) License (https://creativecommons.org/licenses/by/4.0/legalcode). Map base layers were modified in QGIS software version 2.18.

of households in the area), crowding (ratio between the average number of residents per household in the area and the average number of rooms serving dormitory in its area), sanitation (percentage of households with indoor plumbing connected to the global network of water supply) and education (proportion of persons 10 to 14 years old literate) for each Information Zone (IZ), unit of analysis employed in that study. Next, income, slum and subnormal agglomerate indicators were arranged in ascending order while education and sanitation were arranged in descending order considering the average value of each of them in the respective IZ. Then, each indicator received a numerical value started by 1, according to the position assumed in this order and, after that, the values obtained by each one IZ were summed up thus obtaining a score representative of the living conditions of the population of IZ. Subsequently, these scores were sorted in ascending order. Thus, higher scores corresponded to lower living conditions. Next, they were grouped in quartiles ZI corresponding to population segments classified in the following categories of living conditions: high, intermediate, low and very low.

Although for the present study all of these procedures have been adopted, as we used data from 2010 Census that did not have all information required for construction of the slum and crowding indicators, we had to exclude the first one and to build the last one as the "ratio inhabitants per room". The 30 SA were composed by a grouping of CT and they were selected considering socioeconomic and demographic criteria. Thus, the SA had greater uniformity within those in the same stratum of LC and more considerable heterogeneity between the strata.

The Municipal Health Secretariat of Feira de Santana has a wide network of public health services distributed throughout the city's urban territory. As this survey is part of a platform of longitudinal clinical and epidemiological studies on dengue, Zika and chikungunya, we considered important that the population residing in the intra-urban spaces selected for these studies resided in the territories covered by Public Health Units. This way, in the course of the study, the research team would have access to information about the occurrence of cases of these arboviruses. Then, all SA should be close to one Basic Health Unit (BHU) or Family Health Unit (FHU) that offered good quality care for the population. Professionals of Epidemiologic Surveillance of Feira de Santana/FSA city indicated the Basic Health Units (BHU) and the Family Health Units (FHU), which provided better quality care. They selected 46 of the 92 units located in urban FSA, which 14 were BHU. All 14 BHU were considered for the study because they performed complex care, and have health agents on their team. The other 16 SA were selected in CT where the FHU were not located in violent areas and offered good quality care. Health professionals who coordinated these units made this selection. As CT varied greatly in their population size, the SA had minimum 773 and maximum 2,442 people, according to Brazilian Institute of Geography and Statistics(IBGE) data. Then, each SA had one, two or three CT with an average of 1,310 people residing in each SA, making a total of 40 CT.

## Data collect

A SA census was conducted by fieldworkers who visited all households in the selected SA and invited all household members from one-year-old to take part in the study. We excluded those who lived in that household for less than six months. If all residents or any of them were not found at the time of the visit, a second attempt was made at another time or on weekends.

We used a semi-structured questionnaire, installed on tablets, divided in two sections: a) household section: address of the domicile; names of all residents and household position and household income (In 2017, the national minimum annual wage was approximately US$ 300.00); b) individual section: identification data; socio-demographics; self-reported morbidity, specifically related to the three arboviruses that circulate in FSA (Zika, dengue and chikungunya); presence of co-morbidities and continuous use of medication.

## Determination of sample size

The sample design was in a two-stage conglomerate, with the sample fraction of the 1st stage, the sentinel units (f1 = 1.0) that were self-represented, that is, they were all part of the study, and the sample fraction of the 2nd stage, selection of individuals (f2 = f/f1) of 0.05, to compose the sample of each sentinel area, in order to maintain the simple random sampling fraction of 0.05 (f = f1*f2).

To calculate the sample size, we used the simple random sampling formula, with sample fraction f = n/N, considering the prevalence of the specific event of 50%; significance level of 5% and sampling error of 0.025. Thus, the sample size of 1,504 was obtained. This sample was expanded to 2,114 by assuming an effect around 1.42. In addition, we added 40% more in the final sample in order to avoid loss of power due to refusals, resulting in 2,960 individuals who were randomly recruited.

## Blood collect and laboratory procedures

The blood collection started in April 2017 and concluded in September 2017. After signing the specific FICF, trained laboratory technicians collected 5 mL of blood per venipuncture, in accordance with current biosafety standards. Local laboratory staff separated the serum by centrifugation and stored it at –20˚C. We transported the frozen aliquots to the virology laboratory of the Federal University of Bahia/Institute of Health Sciences, where the laboratory tests were processed.

The presence of IgM or IgG antibodies against CHIKV was detected in serum samples using the Chikungunya anti-virus ELISA (IgM and IgG) from Euroimmun (Lübeck, Schleswig-Holstein, Germany), following the manufacturer's instructions [15]. This test is based on a calculation given by the ratio of the coefficient of extinction of the sample and the coefficient of extinction of the calibrator provided in the kit. Samples with ratio below 0.8 are considered negative, while those above or equal to 1.1 are positive. Samples above or equal to 0.8 and below 1.1 are considered as inconclusive.

## Entomological indicator

The entomological indicator used was the Premise Index (PI) [16] obtained for each SA as the percentage of all inspected premises which were found with at least one positive breeding site for *Aedes aegypti* larvae. These data were provided by professionals of Entomological Surveillance System of the municipality.

## Data analysis

We estimated the crude seroprevalence by SA and presented the respective Confidence Intervals of 95%. The global seroprevalence, by LC strata and by sex were estimated and presented the respective Confidence Intervals of 95%, design effect and the intra class correlation (ICC). The Seroprevalence Ratio by LC was estimated as well as the respective Confidence Intervals of 95%. Pearson chi squared ($\chi^2$) tests were used to test association between CHIKV seroprevalence and the four LC strata assuming $p<0.05$ and Poisson Regression was used to estimate the Seroprevalence Ratios, assuming as reference the high LC. To test the correlation between CHIKV seroprevalence and ILC, population density and PI, we applied the non-parametric Spearman correlation test. All analyses were performed using Stata software version 12.0 (Stata Corporation, College Station, USA).

## Results

We visited 15,045 households and 6,977 of these were excluded (being commercial, uninhabited, or refusals). In the remaining 8,068 (53.6%) households, we interviewed 16,620 inhabitants. Next, 2,960 (17.8%) of them were randomly selected to the serosurvey, but only 1,981 (66.9%) agreed to participate (Fig 2), being 67.6% women and almost 25% aged over 60 years. This distribution was similar to that of the interviewed population.

The majority of participants self-declared as mixed race (62.2%) or black (23.0%). However, in the SA with high LC there was a larger proportion of white people (19.2%) than in the low LC (8.8%). The percentage of inhabitants reporting more than 12 years of formal education was highest in the high LC area (11.7%) and decreased as the LC declined to the very low area (3.3%). It was observed that in all SA predominated income between 1 and 2 minimum wages and only in the high and intermediate LC strata there were SA with income greater than five minimum wages (Table 1). These differences were statistically significant. The percentage of data missing related to income variable was high in all AS (average 35.1%), ranging from 29.3% to 38.0% in low and high LC, respectively.

The global CHIKV seroprevalence was 22.1% (CI95% 16.7–28.6), being 82.3% of these were IgG only; 11.2% IgM only; and 6.5% both IgG and IgM and varied from 2.0% to 70.5% in the SA 29 and 12, respectively, both located in the low stratum (Fig 3A and 3B). The median value of seroprevalence in SA of high LC was 9.9% (3.2 to 16.7%), 18.9% (5.4 to 47.7%) in intermediate, 14.1% (2.0 to 70.5%) in low and 12.9% (8.0 to 55.4%) in very low LC (Fig 4). It was 15.1% (CI95%11.6–19.4) for women and 7.0% (CI95% 4.9–9.9) for men. The value of this indicator by LC was 11.4% (CI95% 6.6–19.0) in high, 27.3% (CI95% 17.3–40.2) in intermediate, 20.6% (CI95% 10.5–36.4) low, and 26.9% (CI95% 9.7–55.9) in very low LC. The means of population density varied from 4,878.9 in very low to 5,674.2 inhab/km$^2$ in high LC strata. Regarding to PI, the means varied from 1.58 in very low to 2.74 in high LC strata. The association between the seroprevalence and LC of SA was not significant (Pearson chi squared, p = 0.15). The ICC was estimated in 0.20 (CI95% 0.17–0.23) (Table 2). The Spearman correlation test evidenced independence between LCI and seroprevalence (r = 0,11 and p = 0,57). Fig 5 shows the heterogeneity in the distribution of seroprevalence by LC strata, according to the LCI. The correlations were not significant for all LC strata: high (r = -0.38; p = 0.46), intermediate (r = 0.35; p = 0.30), low (r = -0.04; p = 0.93) and very low (r = -0.50; p = 0.39).

The seroprevalence was higher in ≥40 years old (25.1%) than in < 40 years old (18.0%) and this difference was significant (p< 0.001). The seroprevalence ratios of chikungunya in the intermediate, low and very low LC strata were more than twice that of the high LC stratum. However, this ratio was statistically significant only in intermediate LC (Table 3).

There was negative correlation, no significant (r = -0.18; p = 0.34), between CHIKV seroprevalence and PI, whereas to population density it was positive and significant (r = 0.24; p = 0.02). Even removing the highest seroprevalence, the population density was still associated (r = 0.24 and p = 0.03).

## Discussion

In this population-based study, the global CHIKV seroprevalence of the 30 SA of FSA was 22.1%, ranging from 2.0% to 70.5% among them. This seroprevalence was higher in women and in ≥ 40 years old. There was a weak positive correlation, but significant, between CHIKV seroprevalence and population density and also weak negative correlation, no significant, with PI. The magnitude of seroprevalences of the SA included in each stratum, except in high LC, was very heterogeneous. The Seroprevalence Ratios of the LC strata showed significant

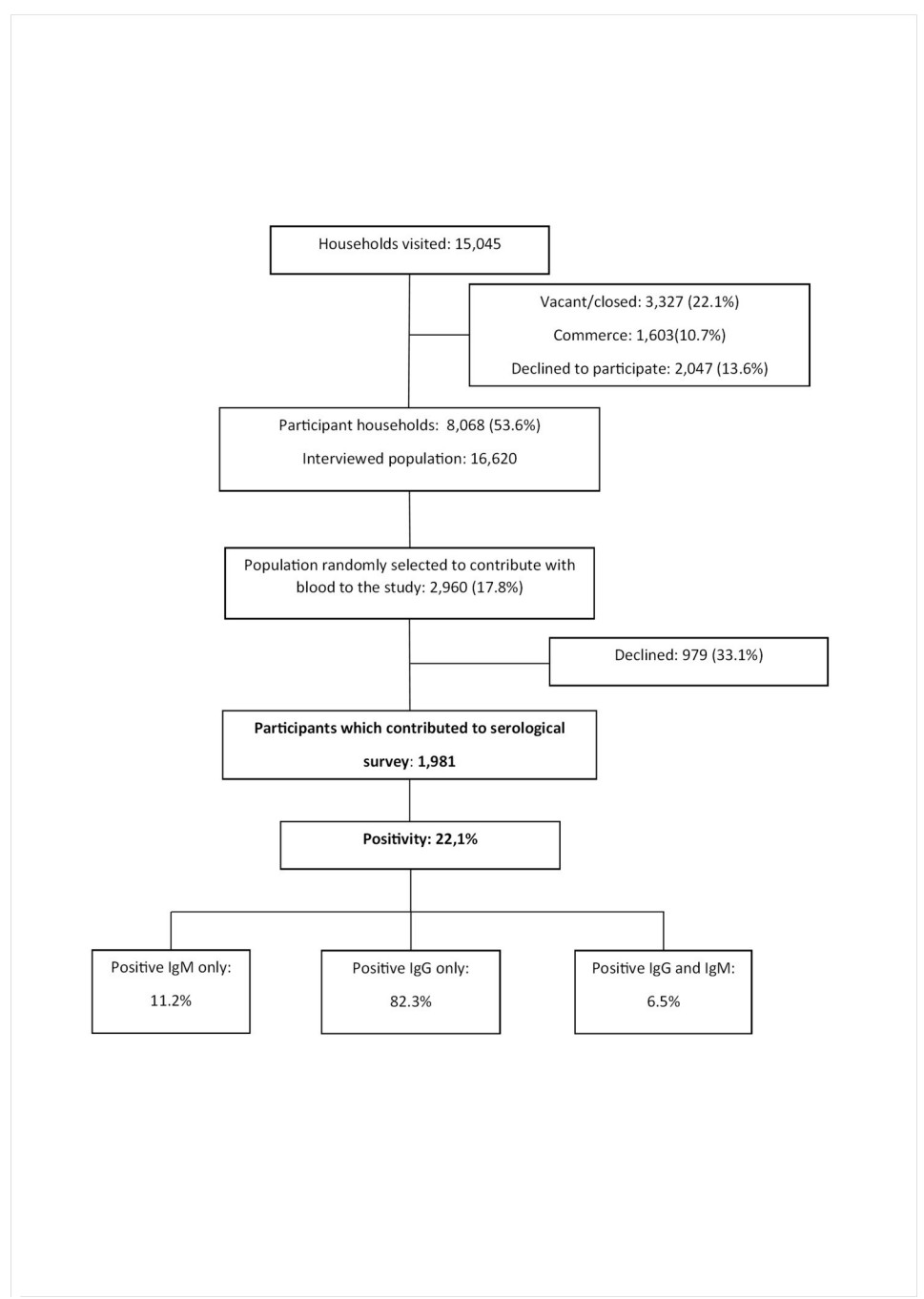

**Fig 2. Flowchart of seroprevalence survey of CHIKV in dwellers from 30 Sentinel Areas of Feira de Santana/FSA.** Bahia-Brazil, 2017.

difference only between high and intermediate LC. The Spearman correlation test evidenced independence between LCI and seroprevalence of SA for all LC strata.

The global seroprevalence in the SA of this study was 2.6 times lower than the 57% observed in the epicentre of the CHIKV epidemic in this same city, in 2015 [17], and it is noteworthy that only in two SA this indicator was higher than 50%. When we compare the results of this study with a seroprevalence survey of dengue, conducted with comparable methodology and a

**Table 1. Sociodemographic characteristics of the population of serological survey for chikungunya by Sentinel Area (SA) and Living Conditions (LC) strata.** Feira de Santana–Bahia, April to September, 2017.

| LC strata | SA number | Serosurvey population (n) | Race (%) | | | | Education in years of study (%) | | | | Household income in NMW[1] (%) | | | | |
|---|---|---|---|---|---|---|---|---|---|---|---|---|---|---|---|
| | | | White | Black | Mixed | Others/ missing | <8 | 8–11 | >12 | Missing | <1 | 1–2 | 2–5 | >5 | Missing |
| High | 6 | 50 | 18.0 | 18.0 | 58.0 | 6.0 | 34.0 | 50.0 | 12.0 | 4.0 | 26.0 | 38.0 | 8.0 | - | 28.0 |
| | 26 | 62 | 19.4 | 25.8 | 54.8 | - | 38.7 | 48.4 | 4.8 | 8.1 | 14.5 | 41.9 | 1.6 | - | 41.9 |
| | 23 | 60 | 31.7 | 3.3 | 65.0 | - | 41.7 | 48.3 | 10.0 | - | 8.3 | 50.0 | 6.7 | - | 35.0 |
| | 3 | 68 | 16.2 | 16.2 | 66.2 | 1.5 | 36.8 | 44.1 | 16.2 | 2.9 | 16.2 | 39.7 | 13.2 | 2.9 | 27.9 |
| | 5 | 41 | 12.2 | 4.9 | 82.9 | - | 48.8 | 43.9 | 7.3 | - | 22.0 | 14.6 | 9.8 | 2.4 | 51.2 |
| | 8 | 71 | 9.9 | 22.5 | 66.2 | 1.4 | 29.6 | 47.9 | 19.7 | 2.8 | 14.1 | 38.0 | 2.8 | 1.4 | 43.7 |
| | **Mean** | **59** | **19.5** | **15.1** | **65.5** | **1.5** | **38.3** | **47.1** | **11.7** | **3.0** | **16.9** | **37.0** | **7.0** | **1.1** | **38.0** |
| Intermediate | 16 | 47 | 12.8 | 38.3 | 48.9 | - | 40.4 | 46.8 | 10.6 | 2.1 | 46.8 | 17.0 | 2.1 | - | 34.0 |
| | 27 | 56 | 19.6 | 33.9 | 42.9 | 3.6 | 44.6 | 48.2 | 7.1 | - | 39.3 | 51.8 | 3.6 | - | 5.4 |
| | 9 | 55 | 9.1 | 29.1 | 54.5 | 7.3 | 32.7 | 45.5 | 20.0 | 1.8 | 7.3 | 61.8 | 12.7 | - | 18.2 |
| | 1 | 118 | 8.5 | 15.3 | 75.4 | 0.8 | 44.1 | 49.2 | 5.1 | 1.7 | 19.5 | 32.2 | 2.5 | - | 45.8 |
| | 14 | 82 | 20.7 | 18.3 | 59.8 | 1.2 | 37.8 | 50.0 | 11.0 | 1.2 | 12.2 | 52.4 | 7.3 | - | 28.0 |
| | 4 | 109 | 16.5 | 13.8 | 68.8 | 0.9 | 37.6 | 52.3 | 4.6 | 5.5 | 25.7 | 43.1 | 2.8 | - | 28.4 |
| | 24 | 65 | 13.8 | 15.4 | 70.8 | - | 50.8 | 27.7 | 13.8 | 7.7 | 16.9 | 40.0 | 6.2 | - | 36.9 |
| | 10 | 74 | 12.2 | 40.5 | 44.6 | 2.7 | 40.5 | 45.9 | 6.8 | 6.8 | 33.8 | 18.9 | 2.7 | 1.4 | 43.2 |
| | 7 | 53 | 9.4 | 18.9 | 71.7 | - | 43.4 | 45.3 | 3.8 | 7.5 | 47.2 | 18.9 | - | 1.9 | 32.1 |
| | 19 | 56 | 10.7 | 21.4 | 64.3 | 3.6 | 37.5 | 41.1 | 17.9 | 3.6 | 19.6 | 30.4 | 5.4 | 1.8 | 42.9 |
| | 2 | 71 | 11.3 | 25.4 | 62.0 | 1.4 | 38.0 | 46.5 | 12.7 | 2.8 | 18.3 | 40.8 | 5.6 | 1.4 | 33.8 |
| | **Mean** | **71** | **13.1** | **24.6** | **60.3** | **2.0** | **40.7** | **45.3** | **10.3** | **3.7** | **26.1** | **37.0** | **4.6** | **0.6** | **31.7** |
| Low | 20 | 67 | 14.9 | 31.3 | 49.3 | 4.5 | 52.2 | 37.3 | 7.5 | 3.0 | 22.4 | 34.3 | 3.0 | - | 40.3 |
| | 28 | 82 | 18.3 | 24.4 | 54.9 | 2.4 | 46.3 | 42.7 | 8.5 | 2.4 | 58.5 | 31.7 | 1.2 | - | 8.5 |
| | 15 | 61 | 6.6 | 37.7 | 52.5 | 3.3 | 70.5 | 24.6 | 3.3 | 1.6 | 67.2 | 18.0 | - | - | 14.8 |
| | 29 | 51 | 25.5 | 17.6 | 52.9 | 3.9 | 41.2 | 41.2 | 17.6 | - | 15.7 | 54.9 | 9.8 | - | 19.6 |
| | 13 | 84 | 9.5 | 10.7 | 79.8 | - | 40.5 | 44.0 | 11.9 | 3.6 | 23.8 | 26.2 | 1.2 | - | 48.8 |
| | 22 | 70 | 7.1 | 34.3 | 57.1 | 1.4 | 45.7 | 32.9 | 17.1 | 4.3 | 35.7 | 27.1 | 1.4 | - | 35.7 |
| | 12 | 61 | 9.8 | 45.9 | 42.6 | 1.6 | 47.5 | 49.2 | - | 3.3 | 41.0 | 29.5 | - | - | 29.5 |
| | 21 | 96 | 8.3 | 29.2 | 59.4 | 3.1 | 43.8 | 44.8 | 1.0 | 10.4 | 28.1 | 32.3 | 2.1 | - | 37.5 |
| | **Mean** | **52** | **12.5** | **28.9** | **56.1** | **2.5** | **48.5** | **39.6** | **8.4** | **3.6** | **36.6** | **31.8** | **2.3** | **-** | **29.3** |
| Very low | 17 | 56 | 5.4 | 19.6 | 73.2 | 1.8 | 71.4 | 21.4 | 1.8 | 5.4 | 32.1 | 26.8 | 1.8 | - | 39.3 |
| | 30 | 70 | 4.3 | 28.6 | 60.0 | 7.1 | 60.0 | 24.3 | 7.1 | 8.6 | 48.6 | 18.6 | - | - | 32.9 |
| | 25 | 36 | 16.7 | 11.1 | 69.4 | 2.8 | 38.9 | 52.8 | 5.6 | 2.8 | 8.3 | 44.4 | 2.8 | - | 44.4 |
| | 18 | 52 | 13.5 | 11.5 | 75.0 | - | 63.5 | 34.6 | 1.9 | - | 44.2 | 42.3 | - | - | 13.5 |
| | 11 | 50 | 4.0 | 24.0 | 68.0 | 4.0 | 40.0 | 58.0 | - | 2.0 | 8.0 | 48.0 | 2.0 | - | 42.0 |
| | **Mean** | **53** | **8.8** | **19.0** | **69.1** | **3.1** | **54.8** | **38.2** | **3.3** | **3.8** | **28.2** | **36.0** | **1.3** | **-** | **34.4** |

(1) National Minimum Wage. One national minimum wage is approximately US$ 300.00.

similar period after its re-emergence (about four years after disease introduction) in another Northeast Brazilian city (Salvador) [10], it was observed that the force of DENV transmission was more than three times higher (68.7% ranging from 16.2% to 97.6%) than CHIKV in FSA (22.1% ranging from 2.0% to 70.5%). It is important to highlight that although the seroprevalence of CHIKV was lower than observed in dengue transmission, chikungunya produces a high burden of chronic cases, mainly arthritis, resulting in a considerable negative impact on health and quality of life [18]. This disease can also contribute to the decompensation of associated comorbidities, that can even lead to death [19,20]. As observed in other studies [17,21],

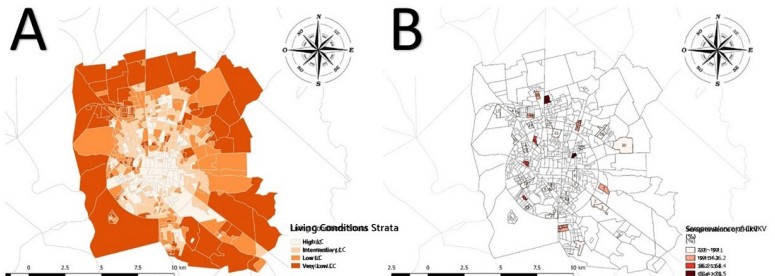

**Fig 3.** A) Map of Census Tract (CT) of Feira de Santana by Living Conditions. B) Seroprevalence of Chikungunya virus by Sentinel Areas (SA). Map base layer was obtained from https://portaldemapas.ibge.gov.br/portal. php#mapa223096 covered by a Creative Commons Attribution 4.0 International (CC BY) License (https:// creativecommons.org/licenses/by/4.0/legalcode). Map base layer was modified in QGIS software version 2.18.

we also found that older women are more affected by these infections. Besides, they are also those who present severe articular manifestations, which usually remain chronic for years [22].

**Fig 4. Seroprevalence of CHIKV by Living Conditions strata in Feira de Santana.** Bahia-Brazil, 2017.

**Table 2. Population density, number of interviewed people, number of serosurvey participants, seroprevalence of chikungunya virus (CHIKV) and Premise Index (PI) by Living Condition (LC) and Sentinel Area (SA).** Feira de Santana–Bahia, April to September, 2017.

| LC | SA number | Population density [1] (inhab/km²) | N Interviewed | % Female interviewed | N Serosurvey | Prev (%) CHIKV (CI95%) | Prev (%) CHIKV female (CI95%) | Prev(%) CHIKV male (CI95%) | PI [2] |
|---|---|---|---|---|---|---|---|---|---|
| **High** | 6 | 5,026.4 | 559 | 59.0 | 50 | 10.0 (4.4–21.4) | 9.1 | 11.8 | 1.73 |
| | 26 | 3,790.0 | 538 | 57.4 | 62 | 3.2 (0.01–11.0) | 0.0 | 10.0 | 3.55 |
| | 23 | 4,665.5 | 892 | 64.6 | 60 | 16.7 (9.3–28.0) | 17.5 | 15.0 | 2.64 |
| | 3 | 7,521.1 | 457 | 63.2 | 68 | 13.2 (7.1–23.3) | 10.4 | 20.0 | 2.11 |
| | 5 | 6,297.9 | 624 | 60.6 | 41 | 9.8 (3.9–22.6) | 11.5 | 6.7 | 2.42 |
| | 8 | 6,744.0 | 470 | 60.6 | 71 | 8.5 (3.9–17.2) | 9.3 | 6.9 | 3.98 |
| | **Mean / Total** | **5,674.2** | **3,540** | **60.9** | **352** | **11.4 (6.6–19.0)** | **11.1 (5.5–21.2)** | **11.9 (7.4–18.4)** | **2.74** |
| | **Deff** | - | - | - | - | **1.07** | **1.11** | **0.40** | |
| **Intermediate** | 16 | 5,133.0 | 274 | 66.4 | 47 | 34.0 (22.2–48.3) | 36.1 | 27.3 | 1.22 |
| | 27 | 4,376.3 | 322 | 67.1 | 56 | 5.4 (1.8–14.6) | 7.9 | 0.0 | 3.55 |
| | 9 | 5,669.1 | 293 | 67.2 | 55 | 12.7 (6.3–24.0) | 16.2 | 5.6 | 1.39 |
| | 1 | 6,672.1 | 490 | 56.3 | 118 | 15.3 (9.9–22.8) | 21.7 | 6.1 | 0.52 |
| | 14 | 6,120.8 | 607 | 58.8 | 82 | 36.6 (27.0–47.4) | 39.7 | 29.2 | 2.89 |
| | 4 | 4,669.0 | 902 | 59.6 | 109 | 47.7 (38.6–57.0) | 47.6 | 47.8 | 1.10 |
| | 24 | 6,801.0 | 802 | 56.1 | 65 | 32.3 (22.2–44.4) | 36.8 | 25.9 | 1.05 |
| | 10 | 2,869.9 | 445 | 65.2 | 74 | 18.9 (11.6–29.3) | 20.4 | 15.0 | 1.57 |
| | 7 | 4,648.9 | 657 | 60.3 | 53 | 11.3 (5.9–22.6) | 8.3 | 16.7 | 1.02 |
| | 19 | 4,133.3 | 897 | 61.8 | 56 | 39.3 (27.6–52.4) | 35.1 | 47.7 | 1.44 |
| | 2 | 7,780.1 | 758 | 58.7 | 71 | 8.5 (3.9–17.2) | 10.9 | 4.0 | 1.68 |
| | **Mean / Total** | **5,352.1** | **6,447** | **61.6** | **786** | **27.3 (17.3–40.2)** | **27.9 (18.5–39.7)** | **26.1 (14.1–43.5)** | **1.58** |
| | Deff | - | - | - | - | 7.23 | 4.28 | 3.66 | |
| **Low** | 20 | 5,967.3 | 631 | 57.8 | 67 | 28.4 (19.0–40.1) | 31.1 | 22.7 | 2.99 |
| | 28 | 4,895.3 | 552 | 73.7 | 82 | 8.5 (4.2–16.6) | 10.8 | 0.0 | 0.78 |
| | 15 | 5,631.6 | 522 | 63.8 | 61 | 31.1 (20.9–43.6) | 29.5 | 35.3 | 2.85 |
| | 29 | 5,865.3 | 476 | 59.9 | 51 | 2.0 (0.01–10.3) | 0.0 | 6.7 | 4.50 |
| | 13 | 3,641.8 | 620 | 56.8 | 84 | 9.5 (4.9–17.7) | 10.7 | 7.1 | 1.10 |
| | 22 | 4,463.9 | 877 | 56.1 | 70 | 18.6 (11.2–29.3) | 15.9 | 23.1 | 1.44 |
| | 12 | 8,452.1 | 311 | 66.9 | 61 | 70.5 (58.11–80.4) | 76.0 | 23.1 | 1.38 |
| | 21 | 5,109.9 | 404 | 67.3 | 96 | 8.3 (4.3–15.6) | 9.1 | 6.7 | 2.19 |
| | **Mean / Total** | **5,503.4** | **4,393** | **62.8** | **572** | **20.6 (10.5–36.4)** | **21.8 (10.2–40.7)** | **18.0 (10.1–29.9)** | **2.15** |
| | Deff | - | - | - | - | 13.90 | 11.89 | 2.21 | |
| **Very low** | 17 | 4,537.9 | 479 | 63.0 | 56 | 55.4 (42.4–67.6) | 55.6 | 55.0 | 2.41 |
| | 30 | 3,255.9 | 519 | 64.0 | 70 | 12.9 (6.9–22.7) | 15.1 | 5.9 | 0.53 |
| | 25 | 6,206.7 | 393 | 58.0 | 36 | 11.1 (4.4–25.3) | 12.5 | 8.3 | 1.40 |
| | 18 | 5,308.4 | 674 | 54.9 | 52 | 28.8 (18.3–42.3) | 27.8 | 29.4 | 2.46 |
| | 11 | 5,085.5 | 175 | 59.4 | 50 | 8.0 (3.2–18.8) | 8.3 | 7.1 | 1.40 |
| | **Mean / Total** | **4,878.9** | **2,240** | **59.9** | **264** | **26.9 (9.7–55.9)** | **27.0 (10.4–54.0)** | **26.73 (8.1–60.3)** | **1.64** |
| | Deff | - | - | - | - | 7.49 | 5.06 | 2.62 | |
| | **Mean / Total** | **5,352.2** | **16,620** | **60.8** | **1,974** | **22.1 (16.7–28.6)** | **15.1 (11.6–19.4)** | **7.0 (4.9–9.9)** | **2,03** |
| | Deff | - | - | - | - | 9.11* | 5.29 | 4.21 | |

1) Population Density estimated from data Brazilian Demographic Census, 2010.

2) Premise Index: % of all inspected premises which were found with at least one positive breeding site for *Aedes aegypti* larvae.

* ICC: 0.20 (CI95% 0.17–0.23)

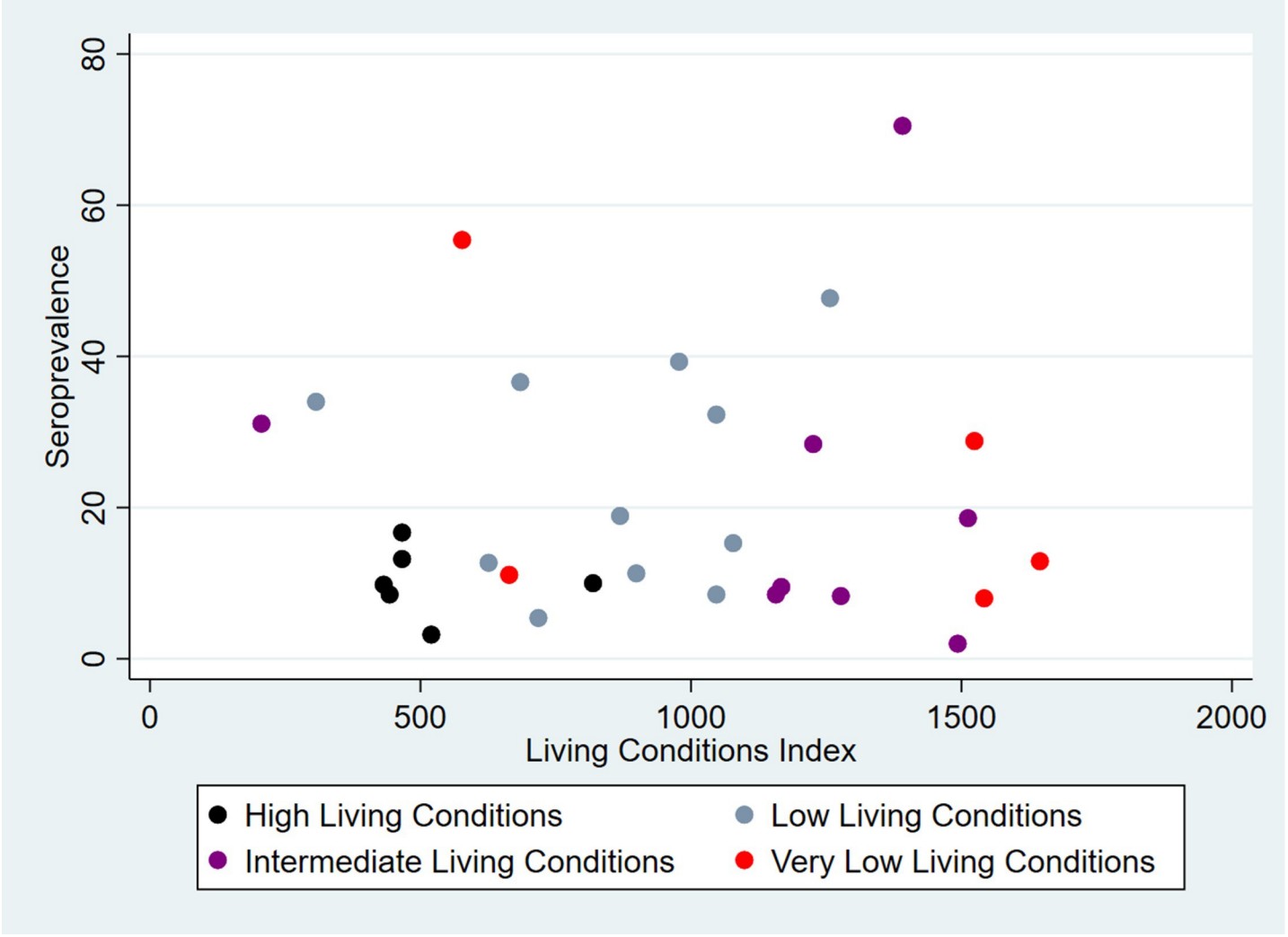

**Fig 5. Scatterplots for the relationship between CHIKV seroprevalence and Living Conditions Index of Sentinel Areas by Living Condition Strata in Feira de Santana.** Bahia-Brazil, 2017.

A finding that stands out in our study is the heterogeneity of the magnitude of CHIKV seroprevalence in intra-urban spaces in the city of FSA, indicating that beyond LC, other factors played a role in the occurrence of this infection, as the human population density that was correlated with CHIKV seroprevalence, although weakly. It was notable that there was only a statistically significant difference between high and intermediate LC strata and no difference was observed for the low and very low LC strata, perhaps due to insufficient "n" sampling. However,

**Table 3. Seroprevalence and seroprevalence ratio of chikungunya by Living Condition (LC) strata.** Feira Santana–Bahia, April to September, 2017.

| LC strata | Seroprevalence | | |
|---|---|---|---|
| | % | Prevalence Ratio | 95% confidence interval |
| High | 11.4 | 1 | - |
| Intermediate | 27.3 | 2.39 | 1.27–4.54 |
| Low | 20.6 | 1.81 | 0.87–3.77 |
| Very low | 26.9 | 2.36 | 0.96–5.81 |

the average seroprevalence of the high LC stratum was much lower than that of all other strata, indicating that possibly LC play an important role in the dynamics of transmission of this arbovirus in intra-urban spaces. In turn, it cannot be disregarded that the epidemiological cycle of CHIKV involves *Aedes aegypti*, a highly competent vector and, consequently, there is a close relationship between agent transmission and the abundance of winged forms of this mosquito in each intra-urban space. Not by chance, conflicting results on this relationship were also found in a systematic review of the relationship between dengue and poverty [23] and specifically to chikungunya, this relation was highlighted as controversial in another study [24].

Furthermore, the only entomological indicator used in the study was the PI, that quantifies only the proportion of households with deposits containing larvae of *Aedes aegypti* in households and peridomiciliary houses. As already mentioned in the literature, it is not a good indicator of viral transmission, as it does not measure the number of adult mosquitoes in the home environment. Some authors also show that there is little evidence of an association between vector indices and dengue transmission that can be safely used to predict outbreaks. On the other hand, it is known that clusters of inadequate houses in overcrowded places with lack of basic sanitation, water storage and accumulation of trash provide an ideal environment for vectors reproduction, mainly *Aedes aegypti* [25]. Usually, these characteristics are common in neighborhoods with lower living conditions.

Although our study has found a weak relationship between the risk of being infected by CHIKV and human population density, it is necessary to remember that, in general, the diseases transmitted by *Aedes aegypti* are also modulated by these factors [26,27]. We observed that the SA of highest CHIKV seroprevalence was a space with crowded residences and substandard slum housing, while the SA with the lowest seroprevalence was an area with mostly single story housing, with vacant lots between the residences. Specifically, the first one has higher population density than the second. This observation is consistent with knowledge about dengue, which studies demonstrated a correlation between the prevalence of DENV infection and population density [10,27,28]. In turn, we also found SA with both relatively high (31.1%—SA 15) and low (2.0%—SA 29) seroprevalence levels with similar population density, 5,631 inhab/km$^2$ and 5,865 inhab/km$^2$, respectively.

These contrasts reveal that the study of the determinants of the occurrence of virus infections transmitted by mosquitoes of *Aedes* genus is very complex, especially in urban centres with co-circulation of three of these agents (DENV, CHIKV and ZIKV), as occur in FSA since 2015 [13]. Possibly it is due to factors inherent to the receptivity of the physical environment [29] and socioeconomic conditions [30,31], beyond other factors involved, such as, the existence of competition between different agents for the vector, the possibility of co-infection in humans and mosquitoes [32] and the vector control measures adopted in each specific space [33].

In the case of FSA, we can hypothesize that the intensification of the vector control measures adopted after the emergence of CHIKV in the city was not universally implemented, but followed the logic of prioritizing the locations that initially produced the majority of cases and that reached the knowledge of public health services. It may have influenced the virus circulation differently during the epidemics in those places that were the target of these actions.

The heterogeneity of the distribution of CHIKV infections can also be observed in intra-urban dissemination of the reported cases of the disease. In fact, the FSA epidemiological bulletins showed that, in 2014, when the CHIKV emerged in this city, two neighbourhoods (epicentres of the disease) concentrated 50% of the reported cases, while the other half was distributed in 74 different neighborhoods [34]. In 2015 and 2016, 11 and 16 neighbourhoods, respectively, accounted for 50% of cases, while the other were distributed in the rest of the 99 neighbourhoods that make up FSA [35–37].

Our study has limitations due to the refusal to participate (22.1%) that can difficult the generalization of the results. As the refusal was mainly of men, it could have contributed to the encounter of greater seroprevalence among women. Besides that, we also did not investigate other factors that modulate the infections by arboviruses, as the availability of habitats for the vector or other environmental factors that influenced the abundance of *Aedes* in the houses; the seroprevalence of other arboviruses in the same time and space that can lead to competition between arboviruses. However, we understand that our investigation is relevant because it showed the transmission of CHIKV in a naive population, in a scenario of co-circulation with more two arboviruses that infect the same vector, in several urban spaces of the city less than three years after the emergence of this agent.

The great heterogeneity in attack rates of CHIKV in studied areas of FSA seems to be a characteristic of its dissemination, since there is a large variability in the seroprevalence levels on surveys available in the literature [8]. Our results indicate that the herd immunity levels and the maintenance of the infestations of the vector have been favouring the CHIKV circulation, either as endemic form or even in the form of new epidemics. In fact, between 2017 and 2019, cases of chikungunya were reported in FSA city with a relatively low incidence (ranging from 21 to 82 cases/100,000 inhabitants). However, in 2020 another epidemic broke out, which already reached an incidence of 889/100,000 inhabitants (up to 45th Epidemiological Week) [38], that is, greater than in 2015, the year of greatest magnitude since the emergence of this agent in FSA.

In summary, despite some limitations of our study, its results indicate that the LC also contribute for the occurrence of CHIKV infections, along with population density. Anyway, it is surprising the no association between the seroprevalence ratios of the low and very low LC and high LC strata, also as well as the failure to find a correlation between vector infestation levels and CHIKV seroprevalence. However, it is already known that the factors that influence the risk of urban arbovirus infections are highly complex at the ecological level.

Controversies and uncertainties still remain, highlighting the need for further studies capable to identify the role of each of the determinants that influence the transmission force of the three agents alone or together, especially in the context of a triple epidemic of urban arboviruses, as occurred in FSA. Finally, it is necessary to seek the improvement of vector control technologies, which must be implemented continuously and universally in urban centres affected by these arboviruses, as well as the improvement of LC, particularly with regard to basic sanitation. Initiatives aimed at the development of drugs against chikungunya, reduction of sequelae and deaths are fundamental and, above all, a vaccine to prevent the occurrence of new infections since the vector control measures that have been applied for many decades have not been very effective.

## Article summary line

There is considerable heterogeneity in the occurrence of CHIKV infections in intra-urban spaces of FSA, being higher in more deprived areas, suggesting that the reduction of the incidence of this disease, until safe and effective vaccines are available, strongly depends on the improvement of the living conditions of the population. This improvement should include, mainly, regular water supply, care with the accumulation of trash, greater access to information, households with less crowding and improve the level of education of population. That is, characteristics that can reduce the exposure to risk factors for CHIKV infection.

## Supporting information

**S1 Database.**
(XLS)

**S2 Strobe Checklist.**
(DOC)

## Author Contributions

**Conceptualization:** Maria Glória Teixeira, Lacita Menezes Skalinski, Enny S. Paixão, Maria da Conceição N. Costa, Florisneide Rodrigues Barreto, Gubio Soares Campos, Silvia Ines Sardi, Rejane Hughes Carvalho, Marcio Natividade, Martha Itaparica, Juarez Pereira Dias, Vanessa Morato, Laura C. Rodrigues, Jimmy Whitworth.

**Formal analysis:** Maria Glória Teixeira, Lacita Menezes Skalinski, Enny S. Paixão, Maria da Conceição N. Costa, Florisneide Rodrigues Barreto, Gubio Soares Campos, Silvia Ines Sardi, Rejane Hughes Carvalho, Marcio Natividade, Martha Itaparica, Juarez Pereira Dias, Soraya Castro Trindade, Bárbara Pereira Teixeira, Vanessa Morato, Eloisa Bahia Santana, Cristina Borges Goes, Neuza Santos de Jesus Silva, Carlos Antonio de Souza Teles Santos, Laura C. Rodrigues, Jimmy Whitworth.

**Funding acquisition:** Maria Glória Teixeira, Lacita Menezes Skalinski, Jimmy Whitworth.

**Investigation:** Maria Glória Teixeira, Lacita Menezes Skalinski, Maria da Conceição N. Costa, Florisneide Rodrigues Barreto, Gubio Soares Campos, Silvia Ines Sardi, Rejane Hughes Carvalho, Marcio Natividade, Martha Itaparica, Juarez Pereira Dias, Soraya Castro Trindade, Bárbara Pereira Teixeira, Vanessa Morato, Eloisa Bahia Santana, Cristina Borges Goes, Neuza Santos de Jesus Silva.

**Methodology:** Maria Glória Teixeira, Lacita Menezes Skalinski, Enny S. Paixão, Maria da Conceição N. Costa, Florisneide Rodrigues Barreto, Gubio Soares Campos, Silvia Ines Sardi, Rejane Hughes Carvalho, Marcio Natividade, Martha Itaparica, Juarez Pereira Dias, Bárbara Pereira Teixeira, Vanessa Morato, Carlos Antonio de Souza Teles Santos, Laura C. Rodrigues, Jimmy Whitworth.

**Project administration:** Maria Glória Teixeira, Lacita Menezes Skalinski.

**Supervision:** Maria Glória Teixeira, Lacita Menezes Skalinski, Jimmy Whitworth.

**Writing – original draft:** Maria Glória Teixeira, Lacita Menezes Skalinski, Maria da Conceição N. Costa, Florisneide Rodrigues Barreto, Gubio Soares Campos, Silvia Ines Sardi, Rejane Hughes Carvalho, Marcio Natividade, Martha Itaparica, Juarez Pereira Dias, Soraya Castro Trindade, Bárbara Pereira Teixeira, Vanessa Morato, Eloisa Bahia Santana, Cristina Borges Goes, Neuza Santos de Jesus Silva, Carlos Antonio de Souza Teles Santos, Laura C. Rodrigues, Jimmy Whitworth.

**Writing – review & editing:** Maria Glória Teixeira, Lacita Menezes Skalinski, Enny S. Paixão, Carlos Antonio de Souza Teles Santos.

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
