## [Decision Letter · Decision Letter 0]

2 Jun 2020

Dear Dra Teixeira,

Thank you very much for submitting your manuscript "Seroprevalence of Chikungunya virus and living conditions in Feira de Santana, Bahia-Brazil" for consideration at PLOS Neglected Tropical Diseases. As with all papers reviewed by the journal, your manuscript was reviewed by members of the editorial board and by several independent reviewers. In light of the reviews (below this email), we would like to invite the resubmission of a significantly-revised version that takes into account the reviewers' comments. 

We cannot make any decision about publication until we have seen the revised manuscript and your response to the reviewers' comments. Your revised manuscript is also likely to be sent to reviewers for further evaluation.

Sincerely,

Marcus VG Lacerda

Associate Editor

Ann Powers

Deputy Editor

Reviewer's Responses to Questions

**Key Review Criteria Required for Acceptance?**

**Methods**

-Are the objectives of the study clearly articulated with a clear testable hypothesis stated?

-Is the study design appropriate to address the stated objectives?

-Is the population clearly described and appropriate for the hypothesis being tested?

-Is the sample size sufficient to ensure adequate power to address the hypothesis being tested?

-Were correct statistical analysis used to support conclusions?

-Are there concerns about ethical or regulatory requirements being met?

Reviewer #1: The figure of CT (Fig.1B) and results of seroprevalence (Fig. 1C) should be put in section RESULTS instead of METHODS, since they are presenting the results of the study.

If the authors are to follow variable of ZI (Paim et al. 2003) and arranged the value of variables in ascending order (income, slum, crowding) and descending order (sanitation, education) and then to determine the LCI using the sum of each variable values, then the order of each variables should all go to the same direction of the living condition. If greater the sum value means higher the ILC, then orders for income, sanitation and education should be ascending, while crowding should be ordered descending, since authors excluded ‘slum’ from the variable list. 

Was any statistical correlation analysis been performed between ILC strata with seroprevalence rate of the CHIKV? (eg. Spearman correlation test).

Reviewer #2: Teixeira et al. conducted a population-based, chikungunya virus (CHKV) seroprevalence study (household interview and serological survey) in Feira de Santana, Bahia-Brazil, an endemic area for Aedes Aegypti transmitted diseases. The design of the study, although not explicitly described, corresponds to a multistage survey based on sampling of clusters (sentinel areas; SA) from strata defined by an adaptation of a previously developed Living Conditions Index. Stratification was designed to minimize intra-stratum and maximize inter-strata heterogeneity of census tracks (CT); however, there is no mention to the procedure used to select clusters (SA) out of strata. Suggestion: Present a more precise and detailed description of the study design and sampling methodology. Field and laboratory procedures are clearly and thoughtfully described. Sample size calculation incorporated the “complex design” nature of the survey in the form of a design effect; however, there is no mention to the parameters used to derive it (i.e., intraclass correlation coefficient) or its resulting value. Suggestion: Include parameters used to estimate the design effect and present its actual value. Finally, the data analysis section does not mention whether the complex design of the survey was incorporated into the point (prevalence) and uncertainty (confidence interval) estimation. Suggestion: Explain how complex design was incorporated in the process of estimation. If it was not the case (i.e., the analysis assumed simple random sampling), then the length of confidence intervals and type I error of statistical tests are underestimated. The study “was approved by Committee on Ethics in Research of Institute of Collective Health/Federal University of Bahia, under number 1.685.039/2016” (lines 226-227).

**Results**

-Does the analysis presented match the analysis plan?

-Are the results clearly and completely presented?

-Are the figures (Tables, Images) of sufficient quality for clarity?

Reviewer #1: It is interesting to see the wide range of the prevalence rate for CHIKV in the city from 2% to 70% with mean only 20.8%. With the presence of SA with prevalence under 5% (2 SAs) and more than 50% (2 SAs), it is good to look with more detail, including the environment, availability of habitats and the dominant type of habitats, along with the entomology index and ILC, since both groups represented extreme polar of CHIKV exposures.

Also since CHIKV is transmitted by Aedes mosquitoes, as dengue viruses, that share the same ecosystem and habits, it will give more insight if we can compare side by side with dengue exposure in the same time periods (whether results of serology survey or record of dengue cases in each SA), whether the picture of both CHIKV and DENV similar in pattern or there is dominance of them that associated with some SAs.

Reviewer #2: Results are presented clearly, following a logical sequence, and in accordance with the analysis plan. These are some general comments/suggestions: 1) Use the decimal point consistently along the manuscript (see line 232 for an example) and 2) when presenting ranges place first the lowest and then the largest value (see lines 119 or 250). More important, please comment on the reason(s) behind not presenting confidence intervals to prevalence estimates. In addition, figure 1 has poor resolution and in table 2 superscripts in parenthesis are not fully explained in the footnote.

**Conclusions**

-Are the conclusions supported by the data presented?

-Are the limitations of analysis clearly described?

-Do the authors discuss how these data can be helpful to advance our understanding of the topic under study?

-Is public health relevance addressed?

Reviewer #1: The conclusion may biased if the authors did not consistent give mark value for each variabel follow the order of the ILC, since the ILC value is the sum of all variables. Putting income and crowding in same order may affect the ILC since income may have positively affect ILC while crowding will affect it negatively.

Low seroprevalence should mean that the area has low exposure of CHIKV, whether it was due to scarce of virus or inability of viruses to transmit due to environment that do not support for vector abundance. No clear analysis on the prevalence rate and entomological indicators. Of course, the low herd immunity will increase the vulnerability for an outbreak in the future. To say that the vulnerability will to be an endemic status, please check the data for at least last three years, if it show that there some cases of CHIKV at each of the year, than it is already endemic, if not than future outbreak and the vector availability can move the area to endemic state.

Authors concluded that risk of infected by CHIKV strongly modulated by population density but did not show any analysis to correlate the population density of SA with the seroprevalence.

Reviewer #2: The conclusions of the study (a low average seroprevalence, an inverse relation of seroprevalence and LC, and the existence of high heterogeneity within strata of LC) are supported by the data; however, I recommend to focus on and develop further the discussion of the main finding of the study in relation to the CHKV seroprevalence estimate (lines 331-333). In relation to the observed high heterogeneity of seroprevalence, I also suggest to deep into the discussion of potential explanations. I am not certain that population density is a factor beyond LC, as this index included crowding. Also, in the study, population density was weakly correlated to seroprevalence and did not seem to explain heterogeneity either (i.e., SA 15 had seroprevalence of 31% and a population density of 5,632 as compared to SA 29 that had seroprevalence of 2% and a population density of 5,865). Please check line 348 for double negative. The authors declare that their study has “limitations” (line 367), however, they only present one in relation to refusals without further development of the implication(s) of such bias (i.e., selection bias) to the results or their generalizability. Finally, the authors did address the public health relevance of their findings in terms of the risk of new epidemics in the area, considering the low herd immunity inferred from an also low seroprevalence.

**Editorial and Data Presentation Modifications?**

Reviewer #1: All figures very poor in resolution and need to be re-draw to increase the clearness and match the required resolution.

Since the title of the manuscript sounds to explore the association of CHIKV seroprevalence with the living condition, it will helpful to present a scatter plot data of each SA versus ILC strata (or even ILC original value, the sum value of all variables).

It also good to present the comparison of seroprevalence rate of each ILC strata in boxplot.

Authors did not consistent in positioning ‘mean’ for each LCI strata. In table 1 it was put at the top of the column while in table 2 was at the bottom of the column for each stratum.

Reviewer #2: N.A.

**Summary and General Comments**

Reviewer #1: Line 67: Prefer to expand LC to be living condition

Line 84: Better to mention completely the name of the city in Summary.

Line 119: better to re-write the number as “0.4% to over 75%”, instead of “over 75% to 0.4%”

Line 146: write the complete name of IBGE or make a citation to reference #14.

Line 169-170: How this range of population size of each SA was determined?

Line 170: …each SA had one, two or three CT (Figure 1C). This is not clear in the Figure.

Line 308-312: Was any statistical correlation analysis been performed between ILC strata with seroprevalence rate of the CHIKV? (eg. Spearman correlation test). 

Line 359-360: Please show the statistical analysis using the study data to support the claim.

Line 369-370: Please back up this statement with statistical test as mention above (line 308-312 and Line 359-360).

Reviewer #2: Teixeira et al. designed and conducted a population-based, chikungunya virus (CHKV) seroprevalence study in Feira de Santana, Bahia-Brazil. The design of the study and the execution of both field and laboratory procedures in such large sample size are some of the main strengths of the survey, although there are missing details in the description of the methods that should be addressed before publication, particularly related to sampling and incorporation of its complexity in the analysis. Perhaps the main weakness of the study is not related to issues of design or execution but the description of the methodology and the discussion of findings. I strongly encourage the authors to accomplish a more exhaustive review of alternative explanations to the observed high intraurban seroprevalence heterogeneity.

PLOS authors have the option to publish the peer review history of their article (what does this mean?). If published, this will include your full peer review and any attached files.

Reviewer #1: Yes: ISRA WAHID

Reviewer #2: No
---

## [Decision Letter · Decision Letter 1]

1 Dec 2020

Dear Dra Teixeira,

Thank you very much for submitting your manuscript "Seroprevalence of Chikungunya virus and living conditions in Feira de Santana, Bahia-Brazil" for consideration at PLOS Neglected Tropical Diseases. As with all papers reviewed by the journal, your manuscript was reviewed by members of the editorial board and by several independent reviewers. The reviewers appreciated the attention to an important topic. Based on the reviews, we are likely to accept this manuscript for publication, providing that you modify the manuscript according to the review recommendations. 

Sincerely,

Marcus Vinícius Guimarães Lacerda

Associate Editor

Ann Powers

Deputy Editor

Reviewer's Responses to Questions

**Key Review Criteria Required for Acceptance?**

**Methods**

-Are the objectives of the study clearly articulated with a clear testable hypothesis stated?

-Is the study design appropriate to address the stated objectives?

-Is the population clearly described and appropriate for the hypothesis being tested?

-Is the sample size sufficient to ensure adequate power to address the hypothesis being tested?

-Were correct statistical analysis used to support conclusions?

-Are there concerns about ethical or regulatory requirements being met?

Reviewer #1: Method section was improved

Reviewer #2: Authors made changes to the section "Areas Selection" in response to one of my comments, however, there is one difference between the text submitted to the "Reviewer's Responses to Questions"(p. 36; "The Municipal Health Secretariat of Feira de Santana has a wide network...") and the corresponding line 181 in the final manuscript: "The Municipal Health Secretariat of Feira de Santana has an extemsive network..." (there is also a typo). In addition, there are two abbreviations (UBS, line 192; IBGE, line 197) not spelled out in the text. The authors did improve the description of sample size calculation and offered an estimate for the "design effect" of 1.42 (line 224), however, I suggest to clarify whether "rho" in the footnote to table 2 corresponds to an intraclass correlation coeffient (ICC), and if so, to include this estimate in the text.

**Results**

-Does the analysis presented match the analysis plan?

-Are the results clearly and completely presented?

-Are the figures (Tables, Images) of sufficient quality for clarity?

Reviewer #1: Results has been improved but interpretation of the results should stated clearly

Reviewer #2: The authors did provide confidence intervals to selected point estimates of prevalence. Please, clarify why in table 2, according to superscript 3, there was no statistically significant difference between high and other living conditions strata, although in table 3 the 95%IC for the prevalence ratio between intermediate and high LC did exclude the unity: 1.27 - 4.54.? Is this because the intermediate, low and very low LC were pooled together as it might be inferred from lines 306-307?. If pooled, what was the reason to do so? I suggest the authors to restrain from conducting correlation analysis, particularly based on a coefficient correlation, to explore associations between seroprevalence and LC, population density or PI. This could be better approached by using regression analysis, for example, Poisson or negative binomial regression of counts with population as exposure or offset variable.

**Conclusions**

-Are the conclusions supported by the data presented?

-Are the limitations of analysis clearly described?

-Do the authors discuss how these data can be helpful to advance our understanding of the topic under study?

-Is public health relevance addressed?

Reviewer #1: From what we see of the fig 4 and 5 and also the correlation test, it was clear that there is no significant correlation of CHIKV seroprevalence and the living condition, it should stated in conclusion or summary explicitly, while suggesting another way or study to measure the risk factors

Reviewer #2: I have no further comments on this regard.

**Editorial and Data Presentation Modifications?**

Reviewer #1: has been suggested in text and example graphs are attached

Reviewer #2: I have no further comments on this regard.

**Summary and General Comments**

Reviewer #1: This was important public health study to show how living condition may influence risk of get CHIKV infection, however the conclusion taken not clearly show how it influence or not.

Reviewer #2: I have no further comments on this regard.

PLOS authors have the option to publish the peer review history of their article (what does this mean?). If published, this will include your full peer review and any attached files.

Reviewer #1: Yes: Isra Wahid

Reviewer #2: No
---

## [Editor Report · Decision Letter 2]

4 Mar 2021

Dear Dra Teixeira,

We are pleased to inform you that your manuscript 'Seroprevalence of Chikungunya virus and living conditions in Feira de Santana, Bahia-Brazil' has been provisionally accepted for publication in PLOS Neglected Tropical Diseases.

Best regards,

Ann M Powers

Deputy Editor

Ann Powers

Deputy Editor

---

## [Editor Report · Acceptance letter]

13 Apr 2021

Dear Dra Teixeira,

We are delighted to inform you that your manuscript, "Seroprevalence of Chikungunya virus and living conditions in Feira de Santana, Bahia-Brazil," has been formally accepted for publication in PLOS Neglected Tropical Diseases.

Best regards,

Shaden Kamhawi

co-Editor-in-Chief

Paul Brindley

co-Editor-in-Chief
